# OPERATOR-CONSISTENT GRAPH NEURAL NETWORKS FOR LEARNING DIFFUSION DYNAMICS ON IRREGULAR MESHES

**Yuelian Li**
Computational and Applied Mathematics
Department of Computer Science
University of Chicago
liyuelian@uchicago.edu

**Andrew Hands**
Department of Computer Science
University of Chicago
hands@uchicago.edu

## ABSTRACT

Learning partial differential equation dynamics on irregular meshes requires preserving the geometric and algebraic structure of the underlying differential operators. In such settings, classical discretizations are difficult to apply and analytical solutions are often unavailable, limiting the applicability of supervision-based learning. We propose an operator-consistent graph neural network with physics-informed constraints for modeling PDE evolution under these conditions. The approach represents the spatial domain as a graph and couples node and edge dynamics through a consistency loss induced by the graph incidence structure, enforcing the discrete gradient divergence relationship during temporal rollout. By preserving operator-level structure, the model improves stability when learning without explicit solution supervision. Experiments on diffusion processes over evolving meshes and real-world scanned surfaces demonstrate improved temporal stability and accuracy compared with graph convolutional and multilayer perceptron baselines, approaching the performance of classical implicit solvers.

## 1  INTRODUCTION

Classical numerical methods(Bengio & LeCun, 2007) solve partial differential equations efficiently on structured grids, but their stability and accuracy often deteriorate outside such settings. Physical processes such as diffusion (Crank, 1975) evolve on complex geometries without mesh regularity or analytical solutions, posing fundamental limitations for learning-based PDE modeling that relies on regular discretizations or access to solution-level supervision. Consequently, learning physical dynamics must rely on geometric structure and physical constraints rather than pointwise solution labels. This motivates the need for learning frameworks that explicitly encode such operator-level structure when modeling diffusion on irregular meshes.

Among learning-based approaches, Graph Neural Networks (GNNs) (Wu et al., 2021) provide a natural representation for irregular meshes, but unconstrained message passing does not guarantee consistency with the differential operators governing the dynamics. When node states and edge-aligned fluxes are learned independently, the resulting dynamics can deviate from physically meaningful behavior during temporal rollout (Sanchez-Gonzalez et al., 2020). Preserving algebraic relationships between discrete operators is therefore essential for stable prediction on unstructured domains.

In this work, we propose an operator-consistent graph neural network with physics-informed constraints for learning diffusion dynamics on irregular meshes. Our contributions are as follows:

- **Graph Neural Network for Diffusion on Irregular Meshes.** We introduce a graph neural network that models diffusion dynamics on irregular meshes by coupling node states with edge-aligned fluxes.
- **Geometry-Aware Operator Constraint via P-Tensor Representations.** We show that enforcing compatibility between node dynamics and edge fluxes corresponds to restricting the learned operator class via first-order P-Tensor representations.

- **Stable Learning on Irregular Meshes.** We demonstrate improved stability and accuracy over evolving and real-world meshes, with performance surpassing GCN and other baseline models and comparable to Crank–Nicolson methods on unstructured geometries.

## 2 Related Work

Recent progress in learning-based PDE solvers has been driven by neural operator methods and graph-based models. Neural operators learn mappings between function spaces and demonstrate strong performance on canonical PDE benchmarks, particularly on structured grids. However, existing neural operator methods are empirically developed and evaluated on uniform Cartesian grids, with limited consideration of irregular mesh discretizations. Meanwhile, Physics-informed graph neural networks provide a flexible alternative by representing spatial discretizations as graphs and integrating graph-based models within traditional numerical time-stepping schemes. In such formulations, geometric structure typically serves as a supporting representation rather than a primary source of inductive bias for modeling dynamics on irregular meshes.

A detailed comparison of existing approaches is provided in Appendix A.

Given that existing approaches are largely developed on structured grids and rely on classical numerical solvers or analytical solutions for supervision, while the structural and physical information available in graph representations is not explicitly utilized, we present a graph neural network framework that operates directly on irregular meshes and incorporates geometric structure and physical information into the modeling of PDE-driven dynamics.

## 3 Operator-Consistent GNN-Based Time-Integration Framework

### 3.1 Graph-Based Representation of PDE States on Irregular Meshes

Let $G = (V, E)$ denote a graph corresponding to a spatial discretization, where the node set $V$ represents mesh vertices and the edge set $E$ represents local adjacency relations induced by spatial geometry. For each $i \in V$, let $x_i$ denote the spatial coordinates of the mesh vertex $i$, and likewise $G$ can be considered to represent the induced geometric connectivity of the mesh. Due to the lack of regular structure, such meshes do not admit standard grid-based discretizations. A scalar field $u_i(t)$ is defined at each node $i \in V$ and evolves over time, with its overall evolution driven by a diffusion-type process.

In classical discrete modeling frameworks(Grady & Polimeni, 2010) defined on simplicial complexes or graphs, spatial geometric structure is typically represented algebraically through an incidence matrix $\mathbf{B} \in \mathbb{R}^{E \times V}$. Within this formulation, applying the incidence matrix to a node-based scalar field, say $x \in \mathbb{R}^V$, yields edge-level quantities corresponding to a discrete gradient, $\nabla x = \mathbf{B}x$, while its transpose, $\mathbf{B}^T = \nabla^T$, induces a discrete divergence acting on edge-level variables. The composition, $\mathcal{L} = \nabla^T \nabla$, of these operators yields a discrete Laplacian. This operator structure induced by the duality between nodes and edges provides a discretization of diffusion processes with a clear geometric interpretation.

Building on this discrete graph-based formulation, we parameterize the underlying spatial operators using a graph neural network. The PDE state is represented as a node-wise scalar field $u_i(t)$ defined on the graph, with node coordinates preserved as geometric inputs and edges constructed from local neighborhood relations. Along each directed edge $(i, j)$, local geometric information is encoded through edge features derived from node coordinates, enabling a geometry-aware parameterization of the underlying discrete spatial operator. Under this representation, we model the PDE as a continuous-time dynamical system evolving on the graph structure, where temporal variation is governed by local operators parameterized by a graph neural network.

At each time $t$, the GNN constructs latent node features $h_i(t)$ and edge features $e_{ij}(t)$ from the current state, node coordinates, and mesh connectivity. Information is propagated through a structured sequence of node-to-edge, edge-to-edge, and edge-to-node message passing operations, yielding a local, geometry-aware parameterization of the underlying discrete spatial operator. Let $F_\theta$ denote a shared neural operator implemented by the GNN. The resulting dynamics are given by

$$\frac{df_i(t)}{dt} = F_\theta(h_i(t), \{\, e_{ij}(t) : j \in \mathcal{N}(i) \,\}),  \tag{1}$$

Temporal evolution of the solution is obtained by applying a standard explicit time integration scheme (e.g., forward Euler) to the learned node-wise dynamics.

## 3.2 GEOMETRY-AWARE PHYSICS-INFORMED OBJECTIVES

To train the proposed model, we adopt a physics-informed objective that incorporates the basic constraints of the underlying diffusion equation. Specifically, the loss function includes a PDE residual loss enforcing the governing equation, together with losses imposing the initial and boundary conditions. These components follow the standard physics-informed loss formulations originally introduced in the PINNs framework (Raissi et al., 2019). Importantly, we do not rely on supervision from classical numerical solvers or known closed-form solutions, and we want to learn dynamics that respect the underlying geometry of irregular meshes. However, although physics-informed training objectives constrain the solution at the level of node-wise PDE states, they do not by themselves guarantee consistency with the discrete geometric operators governing diffusion.

Physically, diffusion describes the transport of particles driven by Brownian motion from regions of high concentration to regions of low concentration (Crank, 1975). Such dynamics depend on how differences between neighboring nodes are transmitted through the local geometric structure. This transmission is characterized by different quantities defined between adjacent nodes, which admit a physical interpretation as diffusive fluxes along edges, capturing both the direction and magnitude of field propagation on the mesh. Because these edge-based fluxes are tightly coupled to geometric adjacency and local spatial structure, they cannot be fully represented at the node level alone.

These observations highlight the limitations of node-wise physics-informed objectives in capturing edge-based transport mechanisms on irregular meshes.

## 3.3 OPERATOR CONSISTENCY VIA P-TENSOR REPRESENTATIONS

To address this gap, we seek an operator-level formulation that captures the coupling between node-wise states and edge-wise quantities in a geometry-aware manner, without committing to a fixed discretization or edge orientation.

This perspective leads naturally to a P-Tensor formulation, a message passing framework introduced by Hands et al. (2024), which characterizes the linear interactions between node- and edge-level representations that are invariant to relabeling of nodes and edges.

### 3.3.1 FIRST-ORDER P-TENSORS ON UNDIRECTED EDGES

**Definition 1** (Directed edge space). *Let $G = (V, E)$ be an undirected graph. We define the set of directed edges as $\overrightarrow{E} := \{(u, v) \mid \{u, v\} \in E\}$. We denote by $\mathbb{R}^{\overrightarrow{E}}$ the vector space of real-valued functions indexed by directed edges of $G$.*

**Definition 2** (First-order P-Tensors on undirected edges). *Let $D = E$ be a set of P-Tensor domains. A first-order P-Tensor layer on $\mathcal{D}$ is a collection*

$$\mathbf{X} = \{\mathbf{X}_e \in \mathbb{R}^e \mid e \in D\}.  \tag{2}$$

*Equivalently, for every $e \in D = E$, $\mathbf{X}_e$ is a real vector indexed by the vertices in the reference domain $e$. We consider the specific points in the space as the set:*

$$\mathcal{D}_1(D) := \{(e, v) \mid v \in e, \ e \in D\}.  \tag{3}$$

*Let $\mathrm{PT}_1(D)$ denote the space of first-order P-Tensor layers on the space of reference domains $D$.*

*Define the map $\phi : \overrightarrow{E} \to D_1(D)$ such that for every $(u, v) \in \overrightarrow{E}$*

$$\phi\big((u, v)\big) = (\{u, v\}, u).  \tag{4}$$

**Proposition 1.** *For all $(u, v) \in \overrightarrow{E}$, $\phi^{-1}\left((\{u, v\}, v)\right) = (v, u)$.*

### 3.3.2 COMPUTING GRADIENTS VIA P-TENSORS

**Definition 3** (Node-wise divergence operator). *Let $\nabla^T : \mathbb{R}^{\overrightarrow{E}} \to \mathbb{R}^V$ s.t. for every $v \in V$ and $g \in \mathbb{R}^{\overrightarrow{E}}$,*

$$(\nabla^T g)(v) = \sum_{u \sim v} (g((u,v)) - g((v,u))) \tag{5}$$
$$= \textit{inflow} - \textit{outflow}.$$

*Let $\rho_\phi \in \mathbb{R}^{V \times \overrightarrow{E}}$ s.t. for every directed edge $e \in \overrightarrow{E}$, $\rho_\phi \delta_e = \delta_{\phi(e)}$. By abuse of notation, we will write $\rho_\phi f = \phi f$, for $f \in \mathbb{R}^{\overrightarrow{E}}$.*

*We define the operator $\phi$ on the space of real vectors indexed on $\overrightarrow{E}$ such that it modifies the indexes. In other words, it is linear, and $\phi(\delta_p) = \delta_{\phi_p}$, and likewise for $\phi^{-1}$.*

**Proposition 2** (A P-Tensor representation of node-wise divergence). *Let $P_\cup$ and $P_\cap$ be the P-Tensor operations that map from first order P-Tensors on E to first order P-Tensors on V such that:*

$$P_\cup(x)_v = \sum_{e \in D_v : v \in e} \sum_{u \in e} x_{e,u}, \qquad P_\cap(x)_v = \sum_{e \in D_v : v \in e} x_{e,v}. \tag{6}$$

*Now, let*

$$\hat{\nabla} := 2P_\cap - P_\cup. \tag{7}$$

*Then for any $g \in \mathbb{R}^{\overrightarrow{E}}$,*

$$\hat{\nabla} \phi g = \nabla^T g. \tag{8}$$

We show in Appendix C.3.4 that requiring consistency with edge-based fluxes can be read as a constraint on the class of node–edge operators, and that the P-Tensor framework provides an equivalent characterization of this constraint at the level of representations.

Such operator-level consistency is not automatically enforced by standard physics-informed losses based solely on PDE residuals. Recent work (Kütük & Yücel, 2025) shown that when certain physical laws are not guaranteed by PDE residuals alone, explicitly enforcing them through additional penalty terms can meaningfully restrict the solution space, by introducing an energy-dissipation penalty to enforce a global physical law. Accordingly, we incorporate the proposed node–edge operator consistency as a soft constraint in the loss function. This term promotes algebraic compatibility between node- and edge-level dynamics through an underlying P-Tensor representation, thereby guiding the model toward a structurally consistent class of discrete operators.

## 4 EXPERIMENTS

We construct irregular-mesh datasets on surface geometries with graph-based connectivity in Appendix B, serving as evaluation benchmarks for physics-informed learning on irregular meshes. We consider diffusion dynamics $\partial_t u = \nabla \cdot (D(x)\nabla u)$ for a scalar field $u(x,t)$ evolving on the surface geometry, and train all models in a physics-informed manner without paired solution supervision.

We reproduced the experimental framework of prior work Zhang et al. (2025) and obtained RMSE in the same $10^{-2}$ range, confirming the correctness of our implementation and training procedure. We evaluate the proposed method against several baselines, including graph convolutional networks (GCN)(Kipf & Welling, 2017), physics-informed neural networks (PINN)(Raissi et al., 2019). Crank–Nicolson (CN)(Crank & Nicolson, 1947) is used to generate deterministic numerical reference solutions under the same discretization for computing MAE and MSE, without implying exact physical ground truth.

As shown in Table 1, our model achieves the lowest final-time errors for diffusion dynamics on irregular meshes. Beyond final-time accuracy, we analyze temporal rollouts, solution stability, and physics consistency. Additional experimental details, further comparisons, training loss/error curves, and ablation studies are reported in the Appendix D.

Table 1: Quantitative comparisons across irregular geometries; GNN denotes our model.

| Dataset / Geometry | GCN MAE ↓ | **GNN MAE ↓** | GCN MSE ↓ | **GNN MSE ↓** |
|---|---|---|---|---|
| Ellipsoid | 3.278e-02 | 2.756e-02 | **6.018e-03** | 8.875e-03 |
| Damage–Healing | 3.196e-01 | 2.013e-02 | 1.229e-01 | **2.033e-03** |
| Stanford Bunny | 7.082e-03 | 5.421e-03 | 6.885e-05 | **2.948e-05** |
| Coconut | 7.918e-03 | 7.393e-03 | **7.387e-05** | 5.774e-05 |
| Leaf | 7.312e-03 | 1.678e-03 | 1.137e-04 | **4.095e-06** |

## 5  MODEL OVERVIEW

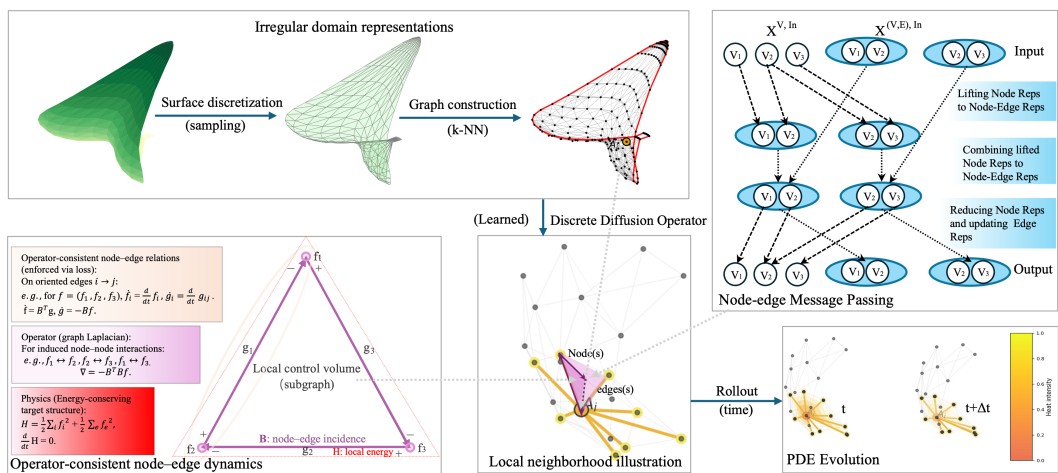

Figure 1: Overview of the the model formulation pipeline, starting from irregular domain representations and a representative local neighborhood on the domain illustrated at $A_i$, we define node–edge representations on which message passing is performed, and interpret the learned node–edge dynamics through an operator-based geometric view, before rolling out in time to obtain PDE evolution.

### ACKNOWLEDGMENTS

We thank Professor Risi Kondor from the University of Chicago Department of Computer Science for his guidance on the GNN aspects of this research and for his valuable suggestions on the dataset, baseline design, experiments, and manuscript preparation. We are also grateful for his development of the P-Tensor framework. Exploring its potential applications in interdisciplinary settings was the original motivation for this research.

We would also like to thank Professor Ahmed Khaled from Northeastern Illinois University for his guidance in building a reproducible irregular mesh dataset, which enabled us to convert real-world surfaces into the irregular meshes required for our experiments.

Furthermore, we would like to thank Professor Koffi Enakoutsa from the University of California, Los Angeles for his guidance on solving PDEs using physics-informed neural networks and for the inspiration behind the PDE residual loss used in our model.

Finally, we want to thank Molly Stoner, the Executive Director of the MPCS program at the University of Chicago, for her warm and continued support throughout the completion of this research.

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

## A  SUPPLEMENTARY DISCUSSION OF RELATED WORK

Table 2: **A structured comparison of learning-based PDE methods and modeling assumptions.**
The comparison highlights systematic differences among existing learning-based approaches to
PDE-induced dynamics and helps identify their respective limitations.

| Method (Refs.) | Representation | Mesh | Time | Node–Edge State | Operator Consistency | Physics | Supervision | Stability |
|---|---|---|---|---|---|---|---|---|
| **PINNs** Raissi et al. (2019) | Continuous | None | Direct | × | × | Residual | × | △ |
| **Graph PINNs** PIGNN Thangamuthu et al. (2022) | Graph | Irregular | Direct | × | × | Residual | × | △ |
| **GNN-based simulators** GNS Sanchez-Gonzalez et al. (2020) | Graph | Irregular | Rollout | × | × | Implicit | ✓ | △ |
| **Hybrid GNN + FEM/FVM** MeshGraphNets (hybrid) Pfaff et al. (2021) Graph–Mesh NN Chenaud et al. (2024) | Graph+Mesh | Irregular | Rollout | △ | △ | Discrete | ✓ | ✓ |
| **Classical Neural Operators** FNO / DeepONet Lu et al. (2021) | Operator | Structured | Direct | × | Implicit | Implicit | × | ✓ |
| **Geometry-Aware Neural Operators** GNO Kovachki et al. (2023) RIGNO Mousavi et al. (2025) | Op.+Graph | Irregular | Direct | × | Implicit | Implicit | × | ✓ |
| **Ours** | Graph | Irregular | Rollout | ✓ | ✓ | Discrete+Struct. | × | ✓ |

✓ Yes / Explicit    △ Partial / Approximate    × No / Not supported

Table 3: **Comparison dimensions for learning-based PDE solvers.** This clarifies the design space
and directly informs the modeling choices and constraints we seek to address.

| Dimension | Description |
|---|---|
| Representation | Model representation assumption determining the solution strategy, e.g., continuous function approximation, graph-based discretization, or operator learning between function spaces. |
| Mesh | Assumed spatial discretization: none (mesh-free), structured grids (e.g., Cartesian lattices), or irregular meshes / point clouds. |
| Time | Temporal handling via direct prediction at queried times or rollout through learned dynamics interpreted as explicit ODE integration (e.g., Euler, RK4). |
| Node/Edge States | Whether latent states are explicitly maintained and updated for both nodes and edges via message passing, rather than edges serving only as connectivity or static features. |
| Operator Consistency | Whether learned updates respect structural properties of the underlying discretized PDE operator (e.g., diffusion/Laplacian structure, flux conservation, symmetry constraints, or consistency under mesh refinement). |
| Physics | How physical constraints are injected, including PDE residuals, conservation laws, and problem-specific conditions such as initial conditions (ICs) and boundary conditions (BCs). |
| Supervision | Training paradigm: reliance on external solution labels versus self-supervision through physics-based constraints. |
| Stability | Empirical stability under long rollouts or repeated application of learned dynamics. |

## B  DATASET CONSTRUCTION AND REPRESENTATION

### B.1  GEOMETRY AND GRAPH REPRESENTATION

The dataset is represented using a unified geometry–graph container. Each mesh sample consists of three-dimensional node coordinates, a graph connectivity structure, and geometry-aware edge attributes. This representation is task-agnostic and independent of the specific partial differential equations (PDEs) applied in subsequent experiments. Graph connectivity is constructed using one of three methods: $k$-nearest-neighbor ($k$-NN) graphs (Belkin & Niyogi, 2003), radius-based neighborhood graphs Gilbert (1961), or intrinsic mesh connectivity derived directly from triangular faces Meyer et al. (2003) when available. Each mesh is also associated with a boundary mask, depending on whether the underlying surface admits an intrinsic boundary or is defined analytically.

### B.2  DATASET CATEGORIES

We consider three categories of spatial discretizations for learning diffusion dynamics on irregular domains: regular meshes, artificially generated irregular meshes, and real-world surface meshes.

**Regular meshes.**  Regular meshes correspond to uniformly discretized grids with fixed stencil connectivity. They admit standard finite-difference discretizations and are known to yield stable numerical schemes under appropriate time-step constraints. In this work, regular meshes are included only as conceptual or numerical references and are not the primary focus of our experiments.

**Analytically generated irregular meshes.**  Analytical irregular meshes are constructed by perturbing or sampling points from analytic geometric domains, such as ellipsoidal surfaces. Irregularity arises from nonuniform point distributions or jitter applied to an otherwise uniform sampling, rather than from intrinsic triangulation. Graph connectivity is constructed explicitly using proximity-based rules, such as $k$-nearest-neighbor or radius-based graphs. Boundary nodes are identified analytically using the defining surface equation within a prescribed tolerance. In our experiments, this category includes physically driven mesh evolution via coupled healing–damage PDEs inspired by phase-field damage models (Bourdin et al., 2008), with stress-informed evolution.

**Real-world surface meshes.**  Real-world irregular meshes are obtained from three-dimensional scans of physical objects, sourced from publicly repositories including Sketchfab and the Stanford University Computer Graphics Laboratory.[1]. These meshes exhibit nonuniform sampling density, varying curvature, and open or partially open boundaries that cannot be reproduced by analytic constructions. When intrinsic triangulation is available, graph connectivity is derived directly from triangle faces, and boundary nodes are identified by detecting edges incident to exactly one face. For closed surfaces without a boundary, diffusion is treated as an initial value problem only.

Table 4: **Summary of meshes, graph construction methods used in our experiments.**

| Dataset | Geometry Source | Mesh Type | Graph Construction | Boundary Source | Surface Type |
|---|---|---|---|---|---|
| Regular grid | Artificial synthetic grid | Regular | Grid stencil | Analytic grid | Open |
| Analytic ellipsoid | Artificial analytic surface | Irregular | $k$-NN/ radius | Analytic surface equation | Open |
| PDE-driven ellipsoid | Artificial analytic surface + PDE evolution | Irregular | $k$-NN | Analytic surface equation | Open |
| Stanford Bunny | Real-world 3D scan | Irregular | $k$-NN | Intrinsic mesh faces | Open |
| Leaf surface | Real-world 3D scan | Irregular | $k$-NN | Intrinsic mesh faces | Open |
| Coconut fragment | Real-world 3D scan | Irregular | $k$-NN | Intrinsic mesh faces | Open |

---

[1]https://sketchfab.com/3d-models, https://graphics.stanford.edu/data/3Dscanrep/

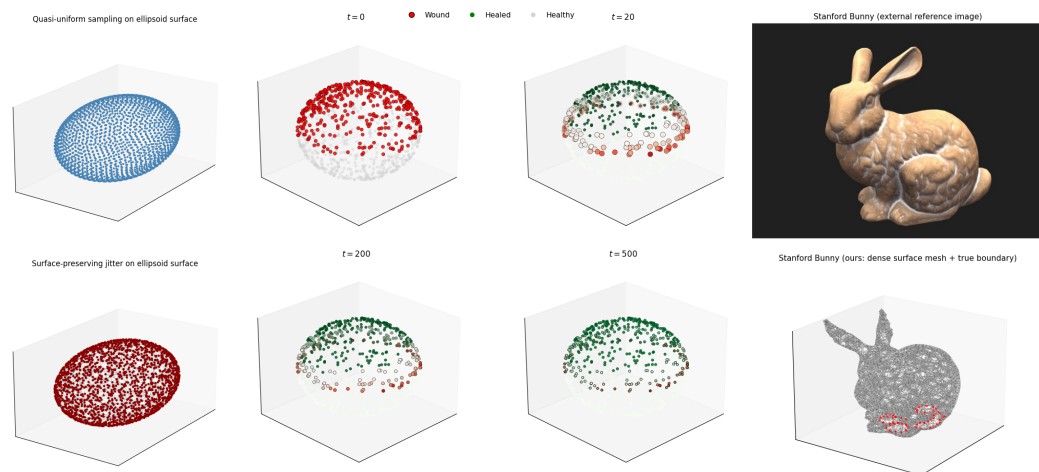

Figure 2: **Irregular surface discretizations and dynamics considered in this work.** *Left:* Synthetic ellipsoid surfaces discretized via quasi-uniform sampling and surface-preserving jitter, providing controlled surface meshes with increasing geometric irregularity. *Middle:* Healing–damage dynamics evolving over time ($t = 0, 20, 200, 500$) on a fixed irregular ellipsoid surface mesh. *Right:* Real-world surface geometry of the Stanford Bunny, shown as an external reference image (top) and our reconstructed dense surface mesh with true topological boundary (bottom).

## C    MODEL FORMULATION

### C.1    PROBLEM CLASS AND DISCRETE DIFFUSION DYNAMICS

We formulate the dynamics at the level of a discrete spatial operator defined on a graph $G = (V, E)$. Let $u(t) \in \mathbb{R}^{|V|}$ denote the vector of node-wise states at time $t$ where $u_i(t)$ corresponds to node $i \in V$. The initial state $u(0) = u_0$ and boundary indicators are assumed to be given.

**Discrete diffusion operator.** Diffusion is modeled through a graph-based discrete Laplacian operator $L$, defined as

$$(Lu)_i = \sum_{j \sim i} w_{ij} (u_i - u_j), \tag{9}$$

where the sum is taken over neighboring nodes $j$ connected to node $i$. The edge weights $w_{ij}$ encode geometric proximity and are chosen as

$$w_{ij} = \frac{1}{\|x_i - x_j\| + \varepsilon}, \tag{10}$$

with a small $\varepsilon > 0$ added for numerical stability. Each node is further associated with a diffusivity coefficient $D_i > 0$, allowing for spatially varying diffusion strength.

**Discrete diffusion dynamics.** The temporal evolution of the system is governed by the discrete diffusion equation

$$\partial_t u_i(t) = D_i (Lu(t))_i, \tag{11}$$

which defines a parabolic diffusion process at the graph level. This formulation does not assume access to any continuous spatial differential operator and is defined purely in terms of the discrete graph structure.

**Relation to continuous diffusion.** The above discrete system can be interpreted as a graph-based discretization of diffusion-type parabolic partial differential equations in the continuum limit. However, no explicit continuous PDE form is assumed or enforced; all operators are defined and learned at the discrete level.

## C.2 LEARNING GRAPH DIFFUSION DYNAMICS UNDER PDE CONSTRAINTS

Our model is a graph neural network that learns discrete-time dynamics on irregular geometries by performing message passing over the graph to estimate the temporal derivative of the node-wise state.

Physical laws are incorporated during training by enforcing residual-based loss functions. Following the discrete diffusion dynamics defined in Eq. (11), training minimizes a collection of residual constraints of the form

$$R(u) = \left(\partial_t u - DLu, \ R_{\mathrm{bc}}(u), \ R_{\mathrm{ic}}(u)\right) \approx \mathbf{0}, \tag{12}$$

where the first term corresponds to the PDE residuals, together with the remaining terms corresponding to a standard physics-informed neural network (PINN) (Raissi et al., 2019) loss function.

Dirichlet boundary conditions are imposed in a weak form by penalizing the solution at boundary nodes, leading to the boundary residual

$$R_{\mathrm{bc}}(u) = u|_{\partial\Omega} - g, \tag{13}$$

where $\partial\Omega$ denotes the true geometric boundary of the input mesh, and $g$ is the prescribed boundary value evaluated on this boundary.

Initial residuals $u_0$ are enforced through

$$R_{\mathrm{ic}}(u) = u(0) - u_0. \tag{14}$$

The overall training objective is given by a weighted sum of these residuals,

$$\mathcal{L} = \lambda_{\mathrm{pde}}\|R_{\mathrm{pde}}\|_2^2 + \lambda_{\mathrm{bc}}\|R_{\mathrm{bc}}\|_2^2 + \lambda_{\mathrm{ic}}\|R_{\mathrm{ic}}\|_2^2. \tag{15}$$

Additional structural constraints can be incorporated through further residual terms, without modifying the underlying graph neural dynamics or geometric discretization.

## C.3 INTERPRETING LEARNED GRAPH DYNAMICS WITH EXPLICIT GEOMETRIC STRUCTURE

In this subsection, we provide an operator-level interpretation of the learned graph dynamics, emphasizing how geometric structure constrains both the model architecture and the induced dynamics.

### C.3.1 FROM GEOMETRY TO GRAPH OPERATORS

Diffusion dynamics on irregular domains are governed by spatial interactions specified through neighborhoods induced by geometric proximity. Representing this structure by graph connectivity provides a discrete approximation of geometric locality, with spatial derivatives captured by relations between adjacent samples.

Adjacent nodes define the discrete support of diffusion, while distance-dependent interactions determine the scale of information propagation. The resulting diffusion operators inherit their structure from the underlying geometry of the domain.

Since diffusion equations depend on local interactions, the discrete diffusion equations introduced in Section C.1 extend to irregular geometries. This formulation defines diffusion-based PDEs on unstructured domains without assuming an explicit continuous spatial differential operator.

### C.3.2 MESSAGE PASSING STRUCTURE ON IRREGULAR MESHES

The model adopts a *Node–Edge–Edge–Node* message passing structure to describe how local field information is transferred and reorganized across the graph. Each propagation cycle starts from node representations that encode scalar field values, which are first lifted to edge space through a node-to-edge transformation.

In this step, geometric relations such as edge length, direction, and angular orientation are incorporated into the edge representation, allowing the model to encode the spatial organization of the irregular mesh. Adjacent edges sharing a common node then exchange information through an edge-to-edge refinement stage, enabling the network to capture local spatial continuity without introducing dense connectivity.

The refined edge states are finally aggregated back to the node level, completing one full pass of information flow from nodes to edges and back to nodes. By stacking multiple such propagation layers, the model progressively integrates geometric and topological information over larger neighborhoods while preserving locality.

### C.3.3   Motivation for operator-consistent node–edge representations

While, Graph Neural Networks provide a message passing structure for learning on irregular domains, in the absence of analytical solutions for supervision, purely data-driven training offers limited constraints on the learned dynamics, which motivates the introduction of additional structural information.

We note that diffusion phenomena obey the principle of energy conservation. Based on this physical fact, we seek to introduce energy-related constraints that are consistent with diffusion into the learning process on discrete graph structures, in order to regulate the learned message passing dynamics.

Therefore, for an arbitrary point $A_i$ on an irregular mesh, within the local neighborhood formed by the adjacent edges, it is not only necessary to perform information transfer at the representation level in order to accomplish a node-wise regression task and predict the solution of the partial differential equation at a given time $t$; it is also necessary to reflect, at the operator level, structures consistent with the underlying diffusion process, so that the learned dynamics can be related to the underlying physical process.

Based on this requirement, we seek a structural representation that simultaneously acts on nodes and edges, serving as a candidate modeling approach to connect node prediction problems with diffusion operator structures.

### C.3.4   Supporting Analysis for the P-Tensor Operator Formulation

This section provides mathematical proofs for the P-tensor constructions introduced in Section 3.3 of the main text. Specifically, we establish the structural properties of P-tensors on node–edge spaces, prove the representation of node-wise divergence in terms of first-order P-tensors, and clarify how skew-coupled node–edge dynamics give rise to energy-conserving diffusion operators. These results collectively justify the use of P-tensor representations as a mechanism for enforcing operator consistency and energy-related constraints in learning diffusion dynamics on irregular graphs.

Firstly, we want to prove a basic property of the edge representation map $\phi$ stated in Proposition 1.

**Claim.** $\phi$ is a bijection.

*Proof.* **Onto:** Let $p \in \mathcal{D}_1(D)$.
By definition, $\exists \{u, v\} \in E(G)$ s.t. $p = (\{u, v\}, v)$.
Since $\{u, v\} \in E(G)$, $(v, u) \in \overrightarrow{E}(G)$, and so $\phi((v, u)) = (\{u, v\}, v) = p$ as desired. Thus, $\phi$ is onto.

**One-to-one:** Let $(u_1, u_2), (v_1, v_2) \in \overrightarrow{E}(G)$ s.t. $\phi((u_1, u_2)) = \phi((v_1, v_2))$. Then,

$$(\{u_1, u_2\}, u_1) = (\{v_1, v_2\}, v_1) \tag{16}$$

which implies for some $u$, $u = u_1 = v_1$ and $\{u_1, u_2\} = \{v_1, v_2\}$. Simplifying the latter gives:

$$\{u, u_2\} = \{u, v_2\}. \tag{17}$$

Since $G$ is simple, and so contains no self loops, we may apply set difference to both sides to get:

$$\{u_2\} = \{v_2\}. \tag{18}$$

This implies for some $v$, $v = u_2 = v_2$. Hence, $(u, v) = (u_1, v_1) = (u_2, v_2)$ as desired.

Therefore, $\phi$ is one-to-one, and so bijective.                                           $\square$

Let $\phi^{-1} : \mathcal{D}_1(D) \to \overrightarrow{E}(G)$ s.t. for all $(\{u, v\}, v) \in \mathcal{D}_1(D)$, $\phi^{-1}((\{u, v\}, v)) = (v, u)$.

Naturally, since $\{u, v\} \in D = E(G)$, this is well defined.

**Claim.** $\phi^{-1}$ is the inverse of $\phi$.

*Proof.* Let $(u,v) \in \overrightarrow{E}(G)$.
Then,

$$\phi^{-1}\left(\phi((u,v))\right) = \phi^{-1}\left((\{u,v\},u)\right)$$
$$= (u,v).$$

This concludes the proof of Proposition 1. $\square$

We next show how the P-tensor operations (Eq. 6) define a mapping from the space of first-order P-tensors on edges to the space of first-order P-tensors on nodes, leading to the edge-to-node divergence operator (Eq. 7), reflecting the difference between incoming and outgoing fluxes (i.e., the net outflow) (Eq. 5) at node $v$, as dictated by a local conservation principle (Jiao, 2022).

*Proof of Proposition 2.*

$$\widehat{\nabla}\phi g = 2P_\cap \phi g - P_\cup \phi g \tag{19}$$

Looking at an arbitrary $v \in V$,

$$= 2P_\cap \delta_{(\{w,h\},w)} - \sum_{e \in D: v \in e} \sum_{u \in e} (\phi g)\left((e,u)\right) \tag{20}$$

$$= 2 \sum_{e \in D: v \in e} (\phi g)\left((e,v)\right) - \sum_{e \in D: v \in e} \sum_{u \in e} (\phi g)\left((e,u)\right) \tag{21}$$

$$= \sum_{e \in D: v \in e} \left(2(\phi g)\left((e,v)\right) - \sum_{u \in e} (\phi g)\left((e,u)\right)\right). \tag{22}$$

Since $G$ is a simple graph and $D = E(G)$, we may write:

$$= \sum_{u \sim v} \left(2(\phi g)\left((\{u,v\},v)\right) - \sum_{w \in \{u,v\}} (\phi g)\left((\{u,v\},w)\right)\right) \tag{23}$$

$$= \sum_{u \sim v} \left(2(\phi g)\left((\{u,v\},v)\right) - (\phi g)\left((\{u,v\},u)\right) - (\phi g)\left((\{u,v\},v)\right)\right) \tag{24}$$

$$= \sum_{u \sim v} \left[(\phi g)\left((\{u,v\},v)\right) - (\phi g)\left((\{u,v\},u)\right)\right]. \tag{25}$$

Applying $\phi^{-1}$, by Definition 3:

$$= \sum_{u \sim v} \left[g\left((v,u)\right) - g\left((u,v)\right)\right] \tag{26}$$

$$= \nabla^T g. \tag{27}$$

$\square$

Finally, we discuss the relation between the edge-to-node divergence operator $(\nabla g)(v)$ induced by the P-tensor construction and the classical graph divergence. To this end, we show that Definition 3 is equivalent to the classical incidence-matrix formulation of graph divergence.

Let $f_t : V \to \mathbb{R}$ be a node field and $g_t : E \to \mathbb{R}$ an edge field. Let $G = (V,E)$ be an undirected graph, where $|V|$ and $|E|$ denote the numbers of nodes and edges, respectively. Fix an arbitrary orientation for each edge i.e. $\mathrm{src}(e) \to \mathrm{dst}(e))$ allows us to identify antisymmetric edge fields on the directed edge set $\overrightarrow{E}$ with functions on the undirected edge set $E$. The resulting edge-by-node incidence matrix is denoted by $B \in \mathbb{R}^{|E| \times |V|}$.

With the incidence matrix $B[e, \text{src}(e)] = +1$ and $B[e, \text{dst}(e)] = -1$, the edge gradient and node divergence are represented as

$$(Bf_t)[e] = f_t(\text{src}(e)) - f_t(\text{dst}(e)), \qquad (B^T g_t)[v] = \sum_{e:\, v=\text{src}(e)} g_t(e) - \sum_{e:\, v=\text{dst}(e)} g_t(e). \quad (28)$$

The quantity $(B^T g_t)[v]$ corresponds to the net outflow at node $v$. We define the node-wise divergence $\nabla^T : \mathbb{R}^E \to \mathbb{R}^V$ by

$$(\nabla^T g_t)[v] := \sum_{u \sim v} g_t(u, v) - \sum_{u \sim v} g_t(v, u), \quad (28)$$

which equals the total inflow minus outflow at node $v$. Accordingly,

$$\nabla^T g_t = -B^T g_t. \quad (29)$$

Here the minus sign reflects the fact that $B^T g_t$ measures the net outflow, whereas $\nabla^T g_t$ measures the net inflow at each node. In other words, the two expressions describe the same edge-to-node mapping, differing only in the choice of flow orientation.

The above discussion establishes the relation between the node-wise divergence in Definition 3 and the classical incidence-matrix formulation. The node-wise divergence itself, however, is defined directly from the P-Tensor representation and the associated tensor operations, without assuming any incidence matrix or fixing an orientation of the graph.

By fixing an arbitrary orientation and introducing the corresponding incidence matrix, one can identify this operator with the classical graph divergence.

This completes the construction of an edge-to-node divergence induced by P-Tensor operations, with the classical graph divergence recovered as a particular representation.

### C.3.5 SUPPORTING ANALYSIS FOR THE INDUCED DYNAMICS AND CONSERVATION PROPERTIES

Let $f(t) : V \to \mathbb{R}$ and $g(t) : E \to \mathbb{R}$ be node and edge fields. Under a fixed ordering of nodes and edges, we represent them as vectors $f(t) \in \mathbb{R}^{|V|}$ and $g(t) \in \mathbb{R}^{|E|}$, respectively. Let $B \in \mathbb{R}^{|E| \times |V|}$ be the incidence matrix defined above. We consider the skew-coupled first-order system

$$\dot{f} = B^T g, \qquad \dot{g} = -Bf. \quad (30)$$

The residuals are defined in the left-minus-right form,

$$R_f = \dot{f} - B^T g, \qquad R_g = \dot{g} - (-Bf) = \dot{g} + Bf. \quad (31)$$

Differentiating the first-order system and substituting the complementary equation yields

$$\ddot{f} = -B^T B f, \qquad \ddot{g} = -BB^T g, \quad (32)$$

from which we identify the node and edge Laplacians

$$L_V := B^T B \in \mathbb{R}^{|V| \times |V|}, \qquad L_E := BB^T \in \mathbb{R}^{|E| \times |E|}. \quad (33)$$

Both are symmetric positive semidefinite, as seen from the quadratic forms, for any $x \in \mathbb{R}^{|V|}$ and $y \in \mathbb{R}^{|E|}$:

$$x^T L_V x = \|Bx\|_2^2 \geq 0, \qquad y^T L_E y = \|B^T y\|_2^2 \geq 0. \quad (34)$$

Define the quadratic Hamiltonian

$$\mathcal{H}(t) := \tfrac{1}{2}\|f(t)\|_2^2 + \tfrac{1}{2}\|g(t)\|_2^2. \quad (35)$$

Differentiating (Eq. 35) and substituting (Eq. (30)) gives

$$\dot{\mathcal{H}} = f^T \dot{f} + g^T \dot{g} = f^T B^T g - g^T B f = 0, \quad (36)$$

so $\mathcal{H}(t)$ is conserved along trajectories of the system.

If one instead enforced $\dot{g} = +Bf$, then

$$\dot{\mathcal{H}} = 2\, g^T B f \neq 0 \quad , \tag{37}$$

which in general breaks the conservation.

Finally, we note that all the above results are independent of the choice of edge orientation. Flipping all edge directions corresponds to replacing $B$ and $g$ by $S_E B$ and $S_E g$, where $S_E \in \mathbb{R}^{|E| \times |E|}$ is a diagonal matrix with entries $\pm 1$. Since $S_E^T S_E = I_E$, this replacement does not change the first-order dynamics. In particular,

$$\dot{f} = (S_E B)^T (S_E g) = B^T g, \qquad \dot{g} = -(S_E B)f = -Bf. \tag{38}$$

It follows that the Laplacians $B^T B$ and $B B^T$ and the Hamiltonian $\mathcal{H}$ are invariant under edge orientation flips. Hence, the above analysis are independent of the chosen edge directions.

## D   EXPERIMENTS AND RESULTS

Through systematic comparisons showing lower errors and more stable training dynamics, this section summarizes the experimental configurations used to produce the results reported in the main paper. All models are implemented in PyTorch and trained using Adam-based optimizers with fixed hyperparameter settings. Additional details on the dataset construction and training procedures are provided in the subsequent appendix sections.

### D.1   EXPERIMENTAL SETUP

**Dataset:**   The experiments are conducted on the Irregular Mesh Dataset introduced in Appendix B. Each surface is represented as a geometric graph with three-dimensional node coordinates, $k$-nearest-neighbor connectivity, geometry-aware edge weights, and explicit boundary node annotations.

**Time Encoding and Rollout:**   Following the continuous-time graph dynamics defined in Section 3.1, time is encoded using a multi-frequency rational Fourier mapping and concatenated with node coordinates and initial states as input features. The model predicts node-wise temporal derivatives together with auxiliary edge dynamics. Time evolution is generated using explicit numerical integration with a forward Euler scheme.

**Baselines:**   We consider two categories of baselines: numerical solvers that provide reference diffusion trajectories, and neural models that learn the same dynamics on irregular meshes.

**Numerical reference.**

1. **CN-irregular**: a Crank–Nicolson solver applied directly to the irregular graph Laplacian with normalized edge weights $w_{ij} = 1/d_{ij}$(Hein et al., 2007), serving as the primary numerical reference.

2. **CN-PDE**: a Crank–Nicolson variant using physically scaled weights $w_{ij} = 1/d_{ij}^2$(Qu & Liang, 2017), corresponding to the continuous diffusion operator and used as a supplementary diagnostic baseline.

**Neural baselines.**

1. **MLP**: a coordinate-based model that predicts node states from spatial locations and initial conditions without exploiting graph connectivity.

2. **CNN**: a convolutional model trained on uniformly discretized grids derived from the same spatial domain, providing a structured-grid reference.

3. **GCN-PINN**: a graph convolutional PINN equipped with Rational Fourier time encoding, which models node-wise diffusion dynamics through message passing on irregular meshes but does not include explicit edge states.

**Evaluations:** All reported errors are computed against the numerical reference (default: CN-irregular) at the final time step $T$, with temporal curves additionally reported to characterize long-term behavior.

Let $u^{\text{pred}}(t) \in \mathbb{R}^N$ and $u^{\text{ref}}(t) \in \mathbb{R}^N$ denote the predicted and reference node states at time $t$, respectively. The discrete Laplacian operator is denoted by $L$, the local diffusivity field by $D = \text{diag}(D_1, \ldots, D_N)$, and the time step size by $\Delta t = T/n_t$.

We report the following error metrics to evaluate prediction accuracy:

**(1) Mean Absolute Error (MAE).** The average absolute deviation at the final time:

$$\text{MAE}(T) = \frac{1}{N} \sum_{i=1}^{N} \left| u_i^{(\text{pred})}(T) - u_i^{(\text{ref})}(T) \right|. \tag{39}$$

This is less sensitive to a small number of nodes with unusually large errors.

**(2) Mean Squared Error (MSE).** The mean squared difference at the final time:

$$\text{MSE}(T) = \frac{1}{N} \sum_{i=1}^{N} \left( u_i^{(\text{pred})}(T) - u_i^{(\text{ref})}(T) \right)^2. \tag{40}$$

MSE emphasizes larger deviations and aligns with the quadratic loss used during training.

**(3) $L_2$ Solution Error.** A normalized $L_2$ error is reported as

$$\|e(T)\|_{2,\text{norm}} = \frac{\left\| u^{(\text{pred})}(T) - u^{(\text{ref})}(T) \right\|_2}{\sqrt{N}}. \tag{41}$$

It allows scale-consistent comparison across meshes of different sizes. The temporal evolution $\|e(t)\|_{2,\text{norm}}$ further reflects the stability of each method over time.

**(4) PDE Residual.** The physics-based residual quantifies the deviation from the diffusion equation:

$$R(t) = \widehat{\partial_t u}(t) - D \mathcal{L} u^{(\text{pred})}(t), \tag{42}$$

where $\widehat{\partial_t u}(t)$ is approximated by a first-order forward difference.

The time-averaged residual magnitude is defined as

$$\|R\|_{\text{time}} = \frac{1}{\sqrt{n_t N}} \left( \sum_{k=0}^{n_t-1} \|R(t_k)\|_2^2 \right)^{1/2}, \tag{43}$$

which summarizes the residual magnitude over the rollout.

**(5) Loss Convergence.** Training and validation losses, including PDE, boundary, and initial condition components, are tracked over epochs on a logarithmic scale, which reveal the stability and convergence behavior of each model.

**(6) Accumulative RMSE (aRMSE).** Following Zhang et al. (2025), with a total time step $N_t$ in period $[0, t]$, the accumulative root-mean-square error over the testing horizon is defined as

$$\text{aRMSE} = \sqrt{\frac{1}{N N_t} \sum_{k=1}^{N_t} \sum_{i=1}^{N} \left\| u_{\text{pred}} - u_{\text{real}} \right\|_2^2}. \tag{44}$$

The aRMSE aggregates prediction errors over all nodes and time steps, reflecting long-term rollout accuracy.

## D.2 EXPERIMENTAL RESULTS

**Correctness Verification Against the PIGNN Reference on 2D Heat Equation.** We evaluate the prediction accuracy using the cumulative RMSE (aRMSE) as in Zhang et al. (2025). Under the same

mesh and PDE settings, our CoreGNN-PINN achieves a final aRMSE of $\mathbf{1.31 \times 10^{-2}}$, compared with $2.09 \times 10^{-2}$ obtained by the GCN-based baseline, while maintaining stable rollout behavior.

Both models are trained on the identical irregular Delaunay mesh with $N = 15681$ nodes and evaluated against the analytic solution of the 2D Heat Equation. The observed error magnitudes are consistent with those reported in the original paper. These results indicate that our implementation is correct and that the proposed model is at least as effective as the reference PIGNNarchitecture under the same physical and discretization settings.

**Comparison 1: 3D Heat Equation on an Irregular Ellipsoidal Point Cloud.**    We first perform a horizontal comparison on a randomly sampled ellipsoidal point cloud, where a 3D Heat Equation is solved on an irregular graph constructed via k-nearest neighbors, and our model is compared with several representative baselines (GCN, PINN, CNN, and MLP) under the same graph Laplacian discretization and temporal integration scheme.

Table 5: Experient on random ellipsoidal point-cloud: horizontal comparison across architectures.

| Model | MAE (CN baseline) | MSE (CN baseline) |
|---|---|---|
| **Our Model** | $\mathbf{2.756 \times 10^{-2}}$ | $8.875 \times 10^{-3}$ |
| GCN(with PINN Loss) | $3.278 \times 10^{-2}$ | $\mathbf{6.018 \times 10^{-3}}$ |
| PINN (MLP + Physics) | $6.342 \times 10^{-2}$ | $1.517 \times 10^{-2}$ |
| CNN | $7.593 \times 10^{-2}$ | $5.525 \times 10^{-2}$ |
| MLP (No Physics) | $1.400 \times 10^{-1}$ | $4.156 \times 10^{-2}$ |

**Comparison 2: Heterogeneous Diffusion PDE Learning on a Damage–Healing Surface Mesh.** In this experiment, we study PDE learning on an irregular surface mesh generated from a time-dependent damage–healing process. The initial field and spatially varying diffusivity are obtained from damage–healing dynamics, while the learning objective enforces a heterogeneous diffusion equation on the mesh. Our model is compared with a GCN-based baseline following the same training and evaluation protocol.

Table 6: Experiment on Damage–healing mesh: comparison of our model and GCN against CN baselines under normalized and PDE-scaled evaluation metrics.

| Metric Type | Model | MAE | MSE | $L_2$ norm |
|---|---|---|---|---|
| CN_norm (normalized) | Our Model | $\mathbf{2.0127 \times 10^{-2}}$ | $\mathbf{2.0330 \times 10^{-3}}$ | $\mathbf{4.5089 \times 10^{-2}}$ |
| CN_norm (normalized) | GCN | $3.1958 \times 10^{-1}$ | $1.2294 \times 10^{-1}$ | $3.5063 \times 10^{-1}$ |
| CN_PDE ($1/d$ weight) | Our Model | $\mathbf{3.4762 \times 10^{-1}}$ | $2.4048 \times 10^{-1}$ | — |
| CN_PDE ($1/d^2$ weight) | GCN | $3.7492 \times 10^{-1}$ | $\mathbf{1.6309 \times 10^{-1}}$ | — |
| **PDE Residual $L_2$** | Our Model | | $\mathbf{5.9005 \times 10^{-2}}$ | |
| **PDE Residual $L_2$** | GCN | | $4.7730 \times 10^{-1}$ | |

**Loss and Physical Consistency Analysis**    Figure 3 compares the training dynamics of our model and the GCN baseline.Our model exhibits stronger oscillations during early training, but converges to a substantially lower steady-state loss, particularly in terms of the PDE residual.

To quantify this behavior, we examine the diagnostic statistics at epoch 200. CoreGNN reports $\dot{f}_{\mathrm{tr}} = 7.58 \times 10^{-2}$, $L_{\mathrm{tr}} = 1.52 \times 10^{-2}$, and a scaling factor $s_{\mathrm{tr}} = 1.36 \times 10^{-4}$, whereas GCN yields $\dot{f}_{\mathrm{tr}} = 4.78 \times 10^{-2}$, $L_{\mathrm{tr}} = 1.84 \times 10^{-2}$, and $s_{\mathrm{tr}} = 3.50 \times 10^{-1}$, where $tr$ denotes statistics averaged over the training rollout trajectory. Although CoreGNN produces a larger raw temporal derivative, the scaled terms become comparable, indicating that the model has adjusted the balance between temporal and spatial components to satisfy the diffusion relation $\dot{f} \approx DLf$. In contrast, GCN maintains a smoother loss trajectory but a larger steady-state imbalance between these terms, reflecting weaker physical regularization.

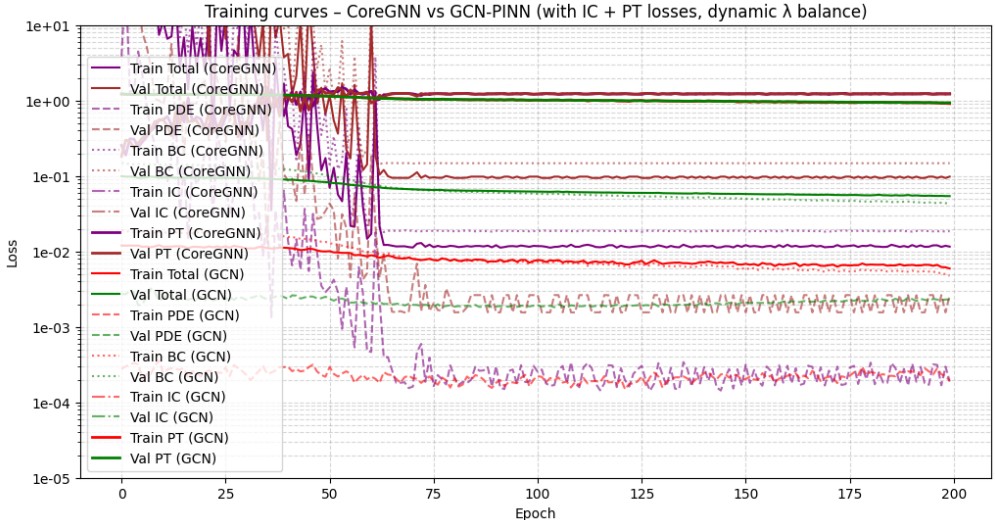

Figure 3: Training and validation loss curves for **CoreGNN_Improved** and **GCN-PINN**, including total, PDE, BC, IC, and P-tensor (PT) losses under dynamic $\lambda$ balancing.

**Comparison 3: Surface Diffusion PDE Learning on Real-world Irregular Meshes.** We evaluate our model on three real-world surface meshes. In all cases, the task is to learn a Surface Diffusion Partial Differential Equation posed on a 2D surface embedded in $\mathbb{R}^3$.

Tables 7–9 show that our model achieves lower errors than the GCN baseline on MAE, MSE, and $L_2$ norm, with smaller PDE residual $L_2$ values. This suggests that our proposed model better learns the dynamics of the surface diffusion PDE on irregular meshes.

We note that under edge-weight $1/d^2$ the error metric is slightly higher than that of the GCN baseline. This is because edge weights in a graph Laplacian are not uniquely defined, but are defined from geometric relations between neighboring vertices (e.g., distance-based weights), emphasizing local mesh structure. Therefore, we retain this edge-weight formulation in our experiments.

Table 7: Experiment on Stanford Bunny Surface: comparison of our model and GCN against CN baselines under normalized and PDE-scaled evaluation metrics.

| Metric Type | Model | MAE | MSE | $L_2$ norm |
|---|---|---|---|---|
| CN_norm (normalized) | Our Model | $\mathbf{5.4211 \times 10^{-3}}$ | $\mathbf{2.9475 \times 10^{-5}}$ | $\mathbf{5.4291 \times 10^{-3}}$ |
| CN_norm (normalized) | GCN | $7.0816 \times 10^{-3}$ | $6.8853 \times 10^{-5}$ | $8.2978 \times 10^{-3}$ |
| CN_PDE ($1/d$ weight) | Our Model | $3.4686 \times 10^{-2}$ | $2.1307 \times 10^{-3}$ | — |
| CN_PDE ($1/d^2$ weight) | GCN | $\mathbf{3.4311 \times 10^{-2}}$ | $\mathbf{2.0299 \times 10^{-3}}$ | — |
| **PDE Residual $L_2$** | Our Model | | $\mathbf{5.4286 \times 10^{-3}}$ | |
| **PDE Residual $L_2$** | GCN | | $9.6463 \times 10^{-3}$ | |

Table 8: Experiment on Coconut inner-surface: comparison of our model and GCN against CN baselines under normalized and PDE-scaled evaluation metrics.

| Metric Type | Model | MAE | MSE | $L_2$ norm |
|---|---|---|---|---|
| CN_norm (normalized) | Our Model | $\mathbf{7.3925 \times 10^{-3}}$ | $\mathbf{5.7742 \times 10^{-5}}$ | $\mathbf{7.5988 \times 10^{-3}}$ |
| CN_norm (normalized) | GCN | $7.9178 \times 10^{-3}$ | $7.3866 \times 10^{-5}$ | $8.5945 \times 10^{-3}$ |
| CN_PDE ($1/d$ weight) | Our Model | $8.2586 \times 10^{-3}$ | $7.4801 \times 10^{-5}$ | — |
| CN_PDE ($1/d^2$ weight) | GCN | $\mathbf{7.1010 \times 10^{-3}}$ | $\mathbf{5.6560 \times 10^{-5}}$ | — |
| **PDE Residual $L_2$** | Our Model | | $\mathbf{7.7321 \times 10^{-3}}$ | |
| **PDE Residual $L_2$** | GCN | | $1.0896 \times 10^{-2}$ | |

Table 9: Experiment on Leaf surface: comparison of our model and GCN against CN baselines under normalized and PDE-scaled evaluation metrics.

| Metric Type | Model | MAE | MSE | $L_2$ norm |
|---|---|---|---|---|
| CN_norm (normalized) | Our Model | $\mathbf{1.6781 \times 10^{-3}}$ | $\mathbf{4.0948 \times 10^{-6}}$ | $\mathbf{2.0236 \times 10^{-3}}$ |
| CN_norm (normalized) | GCN | $7.3119 \times 10^{-3}$ | $1.1372 \times 10^{-4}$ | $1.0664 \times 10^{-2}$ |
| CN_PDE ($1/d$ weight) | Our Model | $\mathbf{1.2425 \times 10^{-1}}$ | $2.9521 \times 10^{-2}$ | — |
| CN_PDE ($1/d^2$ weight) | GCN | $1.2699 \times 10^{-1}$ | $\mathbf{2.9363 \times 10^{-2}}$ | — |
| **PDE Residual** $L_2$ | Our Model | | $\mathbf{2.0355 \times 10^{-3}}$ | |
| **PDE Residual** $L_2$ | GCN | | $1.3135 \times 10^{-2}$ | |

### D.3   ABLATION: HARD ORTHOGONAL CONSTRAINT VS. SOFT CONSISTENCY LOSS.

We conduct an ablation study by replacing the P-tensor consistency loss (penalizing consistency residuals through an additional loss term) with a hard constraint that explicitly projects $\dot{g}$ to be orthogonal to the instantaneous node-induced gradient field $Bf$ at each forward pass. That is to project out the parallel component of $\dot{g}$ via the orthogonal projection $\dot{g} \leftarrow \dot{g} - \frac{\langle \dot{g}, Bf \rangle}{\|Bf\|^2} Bf$, which enforces the hard constraint $\langle \dot{g}, Bf \rangle = 0$ by construction. Compared with the original P-Tensor formulation that penalizes the residuals $\dot{f} - B^T g$ and $\dot{g} + Bf$ from Eq. 30 through a soft loss term, this hard projection enforces exact orthogonality at the output level. As a consequence, the edge dynamics no longer participate in a learnable trade-off between physical consistency and data fitting during optimization, and the strength of the constraint cannot be adaptively adjusted by loss weighting.

Empirically, we find that the hard orthogonality projection does not provide advantages over the soft consistency loss in pure diffusion settings, and performs worse in reaction–diffusion problems such as the Fisher–KPP equation and the Allen–Cahn reaction–diffusion . In particular, training becomes unstable when the reaction term dominates the diffusion term. Under such circumstances, enforcing the diffusion-motivated constraint $\dot{g} \perp Bf$ can over-constrain the dynamics, leading to degraded accuracy and unstable rollouts. By contrast, the soft consistency loss offers a graded regularization that allows controlled constraint violations when necessary, resulting in more robust optimization and slightly better empirical performance.

Table 10: Fisher–KPP reaction–diffusion on the leaf surface: comparison between soft consistency loss and hard orthogonality constraint under different reaction strengths. Denote reaction coefficient by $k$, and PDE residual by $L2$.

| Reaction Regime | Constraint Type | MAE | MSE | $L_2$ |
|---|---|---|---|---|
| Small $k$ | Hard | $6.4519 \times 10^{-2}$ | $2.7766 \times 10^{-2}$ | $1.6663 \times 10^{-1}$ |
| Small $k$ | Soft | $\mathbf{1.6930 \times 10^{-2}}$ | $\mathbf{1.6790 \times 10^{-3}}$ | $\mathbf{4.0975 \times 10^{-2}}$ |
| Large $k$ | Hard | $6.3605 \times 10^{-3}$ | $4.2694 \times 10^{-5}$ | $6.5341 \times 10^{-3}$ |
| Large $k$ | Soft | $\mathbf{1.8981 \times 10^{-2}}$ | $\mathbf{1.8326 \times 10^{-3}}$ | $\mathbf{4.2809 \times 10^{-2}}$ |

Table 11: Allen–Cahn reaction–diffusion on the leaf surface: comparison between soft consistency loss and hard orthogonality constraint under different reaction strengths. Denote reaction coefficient by $k$, and PDE residual by $L2$.

| Reaction Regime | Constraint Type | MAE | MSE | $L_2$ |
|---|---|---|---|---|
| Small $k$ | Hard | $6.2094 \times 10^{-2}$ | $6.4390 \times 10^{-2}$ | $2.5375 \times 10^{-1}$ |
| Small $k$ | Soft | $\mathbf{2.6527 \times 10^{-2}}$ | $\mathbf{3.7854 \times 10^{-3}}$ | $\mathbf{6.1525 \times 10^{-2}}$ |
| Large $k$ | Hard | $7.0374 \times 10^{-2}$ | $3.2208 \times 10^{-2}$ | $1.7947 \times 10^{-1}$ |
| Large $k$ | Soft | $\mathbf{2.3125 \times 10^{-2}}$ | $\mathbf{3.9507 \times 10^{-3}}$ | $\mathbf{6.2855 \times 10^{-2}}$ |

Based on these observations, we adopt the soft consistency loss in all reported experiments, as it provides more stable and robust behavior across different PDE regimes.

