# OpenReview forum: "Operator-Consistent Graph Neural Networks for Learning Diffusion Dynamics on Irregular Meshes"
_ICLR.cc/2026/Workshop/GRaM — ICLR 2026 Workshop GRaM Poster_

### Official Review · Reviewer_6oLJ · 2026-02-22
**Promising operator consistency idea but execution&presentation needs polish**

**Rating:** 6
**Confidence:** 4

**Review:**

This paper proposes a physics informed GNN that enforces operator consistency between node and edge dynamics via a P-Tensor derived consistency loss promoting algebraic compatibility with the discrete gradient divergence structure. Trained without solution supervision the experiments on synthetic and real world surface meshes show improved stability and accuracy over GCN baselines.

1)The core idea for connecting P-Tensor representations to operator consistency in PDE learning is quite creative and fairly non obvious synthesis that bridges two previously separate literatures.The energy conservation analysis (appendix C.3.5) provides solid theoretical grounding.

2)On the Ellipsoid dataset, the proposed model achieves lower MAE (2.756e-02) but higher MSE (6.018e-03 vs 8.875e-03 for GCN, appendix D2, table 5, page 16). This inconsistency is not discussed anywhere and raises questions about node level error distribution.

3)Tables 10 and 11 in Appendix D3 appear to swap labels ("Hard consistency" and "Soft orthogonality") for the large reaction regime. Table 11 also contains a typographical error ("3..7854") and has an identical caption to Table 10. These errors must be fixed.

4)The actual consistency loss term does not appear in the main text so readers must go to the appendix to understand what is being optimized. For a 4page paper this is considerable but for proceedings track, this might significantly affect presentation.

5)Baseline comparison is limited to GCN and MLP. Given the paper avoids solution supervision I would say that a comparison against at least one supervised graph based method would help contextualize absolute accuracy.

Nevertheless, this is a good fit for GRaM, after a few revisions.The paper directly addresses preservation of discrete differential operator structure on irregular geometric domains hence connecting algebraic topology and physics informed learning.

**Pmlr Suitability:**

NA

---

### Official Review · Reviewer_SdDv · 2026-02-22
**A principled and promising GNN for irregular meshes, but manuscript can be improved.**

**Rating:** 5
**Confidence:** 4

**Review:**

## Summary
The paper proposes an operator-consistent graph neural network (CoreGNN) for learning diffusion PDE dynamics on irregular meshes without access to solution-level supervision. The core idea is to couple node and edge dynamics through a consistency loss derived from the graph incidence structure, enforcing a discrete gradient-divergence relationship during temporal rollout. The consistency constraint is motivated through P-Tensor representations [1], which characterize linear node-edge interactions invariant to relabeling. Experiments on synthetic ellipsoids, damage-healing meshes, and real-world scanned surfaces (Stanford Bunny, coconut, leaf) show improvements over GCN [2] and MLP baselines.

## Strengths
-  **Principled operator-level inductive bias:** The insight that node-based PINN methods can't enforce edge-flux consistency is well-motivated and shows a gap in the literature. Tackling this gap by enforcing a consistency constraint in a formal algebraic framework (P-Tensors [1], incidence matrices [4]) gives the approach theoretical credibility beyond ad hoc regularization.
- **High Practical Relevance:** Most neural PDE solvers are still created for structured Cartesian grids or require massive amounts of supervised data from classical solvers. Formulating a self-supervised, physics-informed model that operates directly on complex, real-world irregular geometries addresses a limitation in the field.
- **Supervision-free setting:** Training without paired solution labels on irregular meshes is a practically important and underexplored regime. The paper takes this constraint seriously and designs the whole pipeline around it.
- **Promising Empirical Performance:** The quantitative results presented in Table 1 show improvements over standard GCN approaches on varied, non-trivial topologies, achieving errors closer to classical numerical solvers

## Weaknesses
- **Marginal or inconsistent improvements on real-world meshes:** On the Stanford Bunny and Coconut, the gains over GCN are modest (e.g., Bunny MAE: 5.42e-3 vs. 7.08e-3; Coconut MAE: 7.39e-3 vs. 7.92e-3). On the Coconut under CN-PDE weighting, GCN actually outperforms the proposed model (MSE 5.66e-5 vs. 7.48e-5). The paper does not adequately discuss these inconsistencies or what they imply about the method's reliability.
- **Missing Loss Formulation in Main Text:** The authors introduce the core idea of penalizing node-edge operator consistency via a "soft constraint in the loss function". However, the exact mathematical formulation of this loss term is completely missing from the main text. Section 3.3 sets up the P-Tensor definitions nicely but stops abruptly before showing how these definitions translate into the actual optimizable objective.
- **Limited Baseline Presentation:** Table 1 only compares the proposed GNN against a standard GCN baseline. While the text mentions that PINNs and other methods are used as baselines, none of those comparisons make it into the primary evaluation table in the main text. Manuscript would really benefit from comparing with better well-known baselines.
- **Relying to much on appendix**: I think that in general the manuscript relies to much on the appendix. It might be better to make a full-paper of this submission!

## Questions:
- While your method is designed for irregular meshes, how would it behave on perfectly regular grids. Would your explicit node-edge message passing actually help with physical stability, or would it simply harm efficiency with unnecessary computational overhead compared to a standard CNN?
- The methodology is heavily motivated by and evaluated on diffusion phenomena. How well does this specific P-Tensor consistency framework adapt to systems with dominant advective terms or nonlinear reactions, where transport mechanisms and energy conservation principles behave differently?

**Pmlr Suitability:**

NA

---

### Official Review · Reviewer_hFUC · 2026-02-24
**Not-so-tiny paper with good theoretical backing and promising initial empirical results for diffusion on irregular grids**

**Rating:** 6
**Confidence:** 4

**Review:**

This paper proposes a graph neural network for diffusion on irregular meshes that augments node-state dynamics with edge-aligned fluxes via P-tensor representation and introduces a physics-informed consistency loss. The central idea is to preserve the discrete gradient–divergence relationship during rollouts, thereby constraining the learned operator class and improving stability when training without solution supervision. This is an improvement over just node-based loss functions.

Strengths:
- The paper is generally well-written and well-motivated. PDE learning on irregular grids is applicable in many domains. PDE residual losses do not guarantee compatibility between node states and edge fluxes. The operator-consistency perspective is conceptually sound.
- The training setup avoids paired trajectory supervision and relies on residual losses, which is aligned with PINN-style learning in mesh-free settings.
- The authors present good comparison studies between some baselines and the proposed model.
- The results are supported by a a strong theoretical framework using P-tensors. However it can be improved in the final paper with newer models already mentioned in table 2.

Weaknesses:
- Not an explicit weakness, while t is submitted as a tiny paper, major sections (lit review, method details, results) are in the appendix. I think it is okay with the workshop rules, although reviewers are generally not required to go through appendix. I would suggest polishing it and submitting it to a conference/journal.
- Even with the long appendices, it is missing details on the architecture and hyperparameters of the model. I am also curious about the PI loss weights, which are described as
- While it is stated that "Accordingly, we incorporate the proposed node–edge operator consistency as a soft constraint in the loss function.", there is no explicit loss function defined that combines the PI loss function with this soft constraint. It would be helpful to explicitly state the whole loss function.
- I am sorry if I am missing something fundamental here, but I don't see how eqn 35 is useful for energy conservation with a dissipative diffusion system. Further, these claims "Recent work (K¨ ut¨ uk & Y¨ ucel, 2025) shown that when certain physical
laws are not guaranteed by PDE residuals alone, explicitly enforcing them through additional penalty
terms can meaningfully restrict the solution space, by introducing an energy-dissipation penalty
to enforce a global physical law. Accordingly, we incorporate the proposed node–edge operator
consistency as a soft constraint in the loss function." seems to imply that a energy-dissipation penalty is helpful. The cited paper enforces energy-dissipation, while this work seems to preserve energy conservation. Appendix C also claims energy conservation as a good bias which is not necessarily true for diffusion.
- Benchmarks are mixed, with the GCN outperforming the proposed model in a few cases, especially with the CN_PDE 1/d^2 metric.
- The MSE, MAE and L2 metrics should be supplemented by relative error metrics, and some info about median and max errors. This is especially relevant for cases where the proposed model has better MAE and GCN has better MSE, which suggests there might be large outlier errors with CoreGNN.
- While time rollout error is quantified by a single aRMSE, it would be beneficial to see rollout behavior to demonstrate stability/continuity. Something like plotting MAE/MSE/RMS through time. Does the error increase with time? How does error behave when the model is used on domain beyond training time.
- There is not much information provided about the actual reference PDEs, and fundamental things like time spacing dt.

Questions/suggestions:
- The first line: "Classical numerical methods(Bengio & LeCun, 2007)". The cited chapter is about scaling learning algorithms, and there is no discussion of classical numerical methods in it.
- Did the PDE loss only consist of the loss at that time step or did you have multi-time step loss to enforce continuity and stability? There are also techniques like causal training from the pinn-world which are used to improve stability.
- How well does the model generalize to different pde parameters (including domain residuals and BC/IC)? What if different dt is used?
- Table 1: bold all of column two for consistency
- Table 2: The info about neural operators should be modified.
Neural operators can also be trained in an autoregressive manner (rollout)
I would argue that neural operators are usually trained in a supervised manner. And physics-informed neural operators also exist.
- Table 10, 11
rows three and 4 should be Hard orthogonality, Soft consistency

Small typos:
- space before meanwhile, p small on line 060
- consistent capitalization of graph neural networks
- has shown instead of just shown 183
- two eqn 28 lines 664-674
- two decimal points table 11 row 2, MSE

**Pmlr Suitability:**

NA

---

### Meta-Review · Area_Chair_Kccw · 2026-02-25

**Decision:**

Accept

**Metareview:**

The reviewers agree that the idea of the paper is promising and one remarks explicitly that it is a good fit for GRaM. The paper is reported to be well-motivated with some experiments and baselines, although some are also missing from the main text. The reviewers also agree that the manuscript could be improved in that it relies a lot on the appendix. In general, I don't find this too concerning and since the idea seems well-liked recommend acceptance.

**Relevance To Proceedings:**

Tiny paper — does not apply

**Relevance To Workshop:**

Yes — suitable for GRaM

---

### Decision · Program_Chairs · 2026-03-02

Accept (Poster)